# Deciphering Differential Life Stage Radioinduced Reproductive Decline in *Caenorhabditis elegans* through Lipid Analysis

**DOI:** 10.3390/ijms221910277

**Published:** 2021-09-24

**Authors:** Elizabeth Dufourcq-Sekatcheff, Stephan Cuiné, Yonghua Li-Beisson, Loïc Quevarec, Myriam Richaud, Simon Galas, Sandrine Frelon

**Affiliations:** 1IRSN PSE-ENV Laboratoire de Recherche sur les Effets des Radionucléides sur L’environnement, Cadarache, 13115 Saint Paul-Lez-Durance, France; loic.quevarec@gmail.com; 2Institut de Biosciences et Biotechnologies Aix-Marseille, Aix Marseille University, CEA, CNRS, BIAM, CEA Cadarache, 13108 Saint Paul-Lez-Durance, France; stephan.cuine@cea.fr (S.C.); yonghua.li@cea.fr (Y.L.-B.); 3CNRS, ENSCM, IBMM Université de Montpellier, 34093 Montpellier, France; myriam.richaud@umontpellier.fr (M.R.); simon.galas@umontpellier.fr (S.G.)

**Keywords:** ionizing radiation, oxidative stress, lipids, reproduction, *Caenorhabditis elegans*, chronic exposure, life stage

## Abstract

Wildlife is chronically exposed to various sources of ionizing radiations, both environmental or anthropic, due to nuclear energy use, which can induce several defects in organisms. In invertebrates, reproduction, which directly impacts population dynamics, has been found to be the most radiosensitive endpoint. Understanding the underlying molecular pathways inducing this reproduction decrease can help in predicting the effects at larger scales (i.e., population). In this study, we used a life stage dependent approach in order to better understand the molecular determinants of reproduction decrease in the roundworm *C. elegans*. Worms were chronically exposed to 50 mGy·h^−1^ external gamma ionizing radiations throughout different developmental periods (namely embryogenesis, gametogenesis, and full development). Then, in addition to reproduction parameters, we performed a wide analysis of lipids (different class and fatty acid via FAMES), which are both important signaling molecules for reproduction and molecular targets of oxidative stress. Our results showed that reproductive defects are life stage dependent, that lipids are differently misregulated according to the considered exposure (e.g., upon embryogenesis and full development) and do not fully explain radiation induced reproductive defects. Finally, our results enable us to propose a conceptual model of lipid signaling after radiation stress in which both the soma and the germline participate.

## 1. Introduction

Ionizing radiations (IR), originating from natural radiation and anthropic activities, induce the continuous exposure of wildlife species, requiring the acquisition of specific data to understand its chronic effects and improve its ecological risk assessment. To date, knowledge still needs to be increased since the radiosensitivity of environmental species covers a wide range of dose rates without any explanations [1,2] and risk assessment still uses data coming from acute exposure even though modes of action and response mechanisms can be different than chronic exposure [3,4,5]. Reproduction, which directly impacts population dynamics, is the most radiosensitive individual parameter in many invertebrates [6,7,8,9,10]. Especially in the roundworm *Caenorhabditis elegans*, chronic exposure to IR during its whole development causes a reproduction decrease (upon 50 mGy·h^−1^) [4,11] which is still unexplained fully. The understanding of mechanisms underlying the radioinduced reprotoxic response of *C. elegans* under chronic exposure scenarios is therefore of main concern.

The biological effects of IR are mainly due to oxidative stress which is caused by water radiolysis inducing the production of free radicals and reactive oxygen species (ROS) in cells [12,13]. Studies performed in biological media (e.g., cells) tend to show that in the case of hydroxyl radicals coming from external irradiation, damage to DNA and lipids is a secondary process; proteins are more probable initial targets due to their relative amount and reactivity [14,15,16]. Our first works were thus dedicated to protein studies, in regards to both damage [4] and expression [3]. However, the accumulation of ROS in the organism after prolonged exposures to radiations has also the potential to alter biological pathways [17,18,19], some of which are also involved in the regulation of lipid metabolism [20,21,22,23,24], lipids being molecules that are of main concern for biological functions and signaling. Specifically, IR mediated oxidative stress induces lipid peroxidation and neutral lipid storage in several species [25,26,27], as well as the misregulation of proteins linked to lipid transport (i.e., vitellogenins) in *C. elegans* [3].

In *C. elegans* as in many species, lipid metabolism and reproduction are interconnected [20,28,29,30,31,32,33,34,35]. Thus, each lipid class might have some dedicated roles, signaling, and regulations in various reproduction processes. However, current studies on the effects of IR on lipids have mainly focused on the regulation of total lipid content [25,26,36] although the understanding of compositional changes in lipid classes and fatty acids should give better information on the mechanisms occurring after a stress [37,38].

Additionally, effects on organism reproductive capacity can have several origins including damages on stem cells, gametes, and embryos occurring at different life stages of development [39,40,41,42,43]. In *C. elegans*, life stage sensitivity partly depends on their constitutive cell types (e.g., somatic, germ cells, meiotic, post mitotic). Indeed, somatic and germ cells have different tolerance to radiation damages due to their differences in division rate and repair mechanisms [44,45,46,47,48,49,50,51,52].

To decipher the developmental stage that yields reproduction decline, we identified three critical developmental periods that could be particularly sensitive to radiations:-Embryogenesis (in utero to L1 stage) during which gonad precursors cells are formed;-Early development (until late L3 stage) until the beginning of meiosis during which the gonad cells (somatic and germ cells) divide rapidly;-Gametogenesis (late L3 stage–Young adult (YA)) during which germ cells begin and achieve meiotic divisions.

In our study, *C. elegans* N2 strain, were chronically irradiated in utero until the end of aforementioned stages at the same dose rate of 50 mGy·h^−1^, in order to respect the chronic exposure scenario and dose rate from previous studies [3,11,27]. All of the worms were analyzed at the L4-YA stage. Reproduction parameters, class of lipids, and fatty acids (FA) were assessed for each condition.

## 2. Results

### 2.1. Chronic Radio-Induced Reprotoxic Effects Are Life-Stage Dependent

In a first step, global reproduction endpoints such as brood size and hatching success were assessed at the L4-YA stage. Figure 1A shows a brood size decrease for all scenarios compared to controls (−35% for SC3, *p* = 2.65 × 10^−13^, −11% for SC1, *p* = 0.0195, −12% for SC2, *p* = 0.0130), with an increased effect for SC3 compared to SC1 and SC2 (*p* < 0.001–Tukey contrasts with Bonferroni correction). On the contrary, no significant differences between scenarios were found on the hatching success (Appendix A). To go further, egg laying rate (Figure 1B) and spermatid numbers (Figure 2) were assessed in our conditions. Figure 1B shows a decrease in egg-laying rate for SC3 (*p* = 2.45 × 10^−5^) and SC2 (*p* = 0.0104) compared to controls, and between SC3 and SC1 (*p* = 0.0105–Tukey contrasts with Bonferroni correction). Figure 2A shows a decrease in mean spermatid number compared to controls only for SC3 (−23%, *p* < 1 × 10^−5^). These results altogether mean either an amplification in SC3 of an effect occurring during early development (SC1 and SC2), or two distinct mechanisms, early and late responses causing a different decrease in brood size.

### 2.2. Lipid Classes Are Differentially Modulated According to the Irradiated Life Stage

In *C. elegans*, only 7 to 20% of fatty acids (FA) are de novo synthesized, while the remaining are derived from bacterial food with a daily renewal of phospholipids [28]. In order to discriminate indirect effects coming from the irradiation of dead bacteria lipids’ vs direct effects on nematodes, lipid content in nematodes fed with irradiated OP50 (OP50(i)) was also measured (see Figure A1).

Results of neutral lipids (triacylglycerols-TAG) and polar lipids (phosphatidylethanolamine-PE and phosphatidylcholine-PC) content in the worms are given in Figure 3.

Neutral lipid analysis (Figure 3A) shows a significant increase in TAG content for worms fed on irradiated bacteria compared to controls (*p* = 0.033). Thus, there is an effect of IR on bacteria inducing an increase of TAG content in nematodes. In contrast, there is a significant decrease in TAG content in SC3 compared to OP50(i) (*p* = 0.00062), for which bacteria were irradiated in the same conditions as SC3. However, there is a significant increase in TAGs for SC1 compared to controls (*p* = 0.0021), SC2 (*p* = 0.018) and SC3 (*p* = 4.28 × 10^−5^), SC1 being more comparable to controls regarding the duration of exposure of bacteria (Appendix A). Therefore, there is a significant increase in TAG content for SC1 but a decrease for SC3.

Polar lipids analysis (Figure 3B,C) showed a decrease in PC content for OP50(i) compared to controls (significant with an α risk of 10%, *p* = 0.053), suggesting an effect of irradiated bacteria inducing a decrease in PC content in nematodes. In contrast, there is an increase in PC content for SC3 compared to OP50(i) (*p* = 0.0077) and SC1 (*p* = 0.031). Therefore, there is an increase in PC content for SC3, and there seems to be a decrease in SC1 (compared to controls), however not significant. No significant difference was found for PE content.

Overall, spearman correlation analysis showed a negative correlation between TAG and PC (ρ = −0.47, *p* < 0.05).

### 2.3. FAMEs Analysis Shows the Radiation Induced Modulation of Specific Fatty Acids

Figure 4 describes the fatty acids methyl esters (FAMEs) content from C12 to C22, weighted by the number of proteins, for all our conditions and shows a modulation of some specific fatty acids (FA) mostly for SC3 (PCA Appendix A).

In SC3, C16:1 compared to controls (*p* = 0.0399), C16:1 (9) compared to OP50(i) (*p* = 0.0303) and C17 compared to SC2 (*p* = 0.0114) were increased. While C17Δ (*p* = 0.0034) and C19Δ (*p* = 0.0047) seem to be decreased compared to OP50(i). In SC2, C12 (*p* = 0.0263) and C14 (*p* = 0.0147) decreased significantly compared to OP50(i). Spearman correlation analysis between total dose for each scenario (see Appendix A) and FAMEs content (Appendix A) showed a significant negative correlation for C17Δ, C19Δ, C19, 3-hydroxy C18 with the total irradiation dose and significant positive correlation for C16:1, C16:1(9) and C12.

Among those, we focused on the decrease of cyclopropane fatty acids (CFA) C17Δ and C19Δ in SC3. Since these FA come from bacterial food and are considered as non-essential to *C. elegans* metabolism [28,53,54], their decrease could be due to a defect in food uptake. We tested this hypothesis through: (i) the attraction behavior of irradiated worms towards the bacterial lawn; (ii) their locomotion (thrashing assay); and (iii) the content of ingested and assimilated bacteria in the intestine (see Appendix A and Methods). No effect of irradiation was found on locomotion nor attraction towards bacteria (Appendix A). However, the measurement of bacteria content in the intestine showed that irradiated worms tend to have a faster assimilation rate, albeit not significantly (Appendix A; *p* = 0.07). Then, the decreasing rate of these two CFA in SC3 seems to be linked to their use.

### 2.4. In Situ Changes of Lipid Morphology

To confirm the results obtained after lipid analysis, we investigated the relative area of lipid droplets (LD) in irradiated worms (SC3) compared to controls, through TEM images. We observed that irradiated worm tissues tend to be less resistant to the slicing procedure (red arrows in Figure 5A) which is usually a result of an increased fat content. The measure of LD surface (Figure 5B) showed a significant higher lipid droplet size in irradiated nematodes (*p* < 2 × 10^−16^).

## 3. Discussion

Understanding the molecular determinants of chronic radiation response is of fundamental interest to predict effects at individual level. By studying the effects at different life stages and the changes in lipid composition we were able to better decipher the timing of occurrence and nature of these mechanisms.

### 3.1. Modulation of Lipid Content under Stress: Two Opposite Mechanisms Controlled by the Germline and/or the Soma?

Analysis of lipid class revealed a negative correlation between TAG and PC levels and at different life stages (Figure 6). After exposure throughout embryogenesis (SC1), nematodes showed an increase in TAG content, the major form of long-term energy storage, as is also found in germline-less or germline-deficient mutants [20]. During embryogenesis, only germ cell precursors are present and in a lesser amount than somatic cells. Somatic cells have been shown to have a better repair mechanism than germ cells and thus tolerate higher levels of IR than germ cells [45,46,47]. We suggest that under the SC1 scenario, somatic cells are not affected by IR due to the relatively low total dose or time of exposure (see Appendix A), but germ cells precursors are, concordant with the TAG increase also observed in germline-less *glp-1* mutants. This could partly explain the 11% brood size decrease that we observe in SC1.

On the contrary, nematodes exposed throughout whole development (SC3) showed a decrease in TAG content, consistent with other studies showing an increase of lipid catabolism and autophagy [19,22,48] in *C. elegans* exposed to oxidative stress [20,55], and with previous results in the same condition of exposure for germline-less *glp-1* mutants [27]. Lipid catabolism is controlled mainly by the soma [19] which suggests that somatic cells are affected in the SC3 scenario. In addition to this TAG decrease, we observed a trend of a faster food assimilation rate in the intestine which is concordant with previous findings through an energy-based modeling approach (DEBtox), suggesting that gamma radiation induces an increase in costs for somatic growth and maintenance in *C. elegans* [8]. This could translate to a faster assimilation of nutrients, resulting in the reduced TAG content and specific FA (CFA).

While TAGs decreased in SC3, nematodes showed an increase in PC content. Phospholipids are the main components of cellular membranes and yolk granules [28] composed at 85% of proteins and 15% of lipids. Yolk granules are synthetized in the intestine and transported in the germline through vitellogenins, for which we previously found an overexpression under the same conditions of exposure [3]. According to the literature, a deficiency in germline capacity to uptake yolk granules leads to an accumulation of those in the soma [29,56], and stressed or aging nematodes tend to consume their own intestine to enhance capacity for yolk production [22], which concurs with our present findings.

With these results, we suggest that IR induce an increase in lipid levels (PC and TAG) through germline signals, and a lipid catabolism (TAG) through somatic signals, leading to an unchanged global lipid level. Therefore, at least two opposite mechanisms exist in our conditions and contribute to balance total lipid content. Many studies have been addressing the issue of this double response to oxidative stress [20,27,29,47], and its regulation by the germline [33,55,57], especially regarding the longevity endpoint [33]. However, rather than lipid level, it is the structural properties of FA such as chain length and saturation degree that may contribute to modulate processes such as longevity and reproduction [37,38].

### 3.2. Facing Radiation «All Fat Are Not Equal»: Quality over Quantity

Our results showed significant differences in FAMEs mainly for nematodes exposed throughout their whole development (SC3). In particular, two C16 MUFA were increased, especially the palmitoleic acid (C16:1 (9)). Consistent with this finding, we also observed a decrease of CFA (C17Δ and C19Δ) which was previously reported to be linked to an increase in MUFAs [58]. MUFAs enhance membrane fluidity, reduce oxidative stress and contribute to energy storage and the activation of specific pathways [55] and is often associated to a longer lifespan [37,59], especially in germline-less mutants [60]. However, no difference in lifespan was shown for the N2 strain in our conditions [8,27] and, in the same conditions, germline-less *glp-1* mutants have even shown a decrease in longevity [27], suggesting once again two opposite balanced mechanisms responding to IR.

### 3.3. Reproduction Decrease: Deciphering Mechanisms through Lipid Analysis

From lipid analysis, we suggest that the reduced brood size observed for nematodes exposed during embryogenesis (SC1) could be due to early unrepaired or misrepaired damages on germ cell precursors. Meanwhile, effects occurring in SC3 (i.e., more decreased brood size, egg-laying rate and sperm number decrease) are even more complex to analyze since they occur concomitantly with different phenomena highlighted through our previous studies. This enables one to draw a global map of the results including increased PC content and overexpression of vitellogenins [22], TAG catabolism, MUFA increase, CFA decrease and faster assimilation rate. The coherence of these data is presented in the following part, keeping the objective of explaining the reproductive default after full-development irradiation.

In *C. elegans*, as in many species, egg quality and embryogenesis are conditioned by the activity of vitellogenins [56,61,62,63] and were previously shown to be linked with LD size and their coalescence in the rainbow trout after IR [64]. In our study we found an increase in LD size and C17Δ decrease with an upregulation of vitellogenins in irradiated worms (SC3). Interestingly, these three parameters are all linked to *daf-2* expression [65], involved in the Insulin-like signaling (IIS) pathway, linked to an increase in longevity, resistance to oxidative stress [21] and likely to be involved in the response to IR in our conditions [22,54]. In particular, the upregulation of vitellogenins through the DAF-2/IIS pathway, can result from a sperm dependent signal [23], which is concordant with the observed sperm decrease in our conditions.

In *C. elegans* hermaphrodites, many processes linked to oocytes such as meiotic maturation and ovulation are controlled by the MSP (major sperm protein) found in sperm [66,67,68]. Therefore, a decrease in sperm number can directly affect the egg-laying rate of our experiment. However, another cause of egg-laying rate decrease could be an increase of apoptosis [69] since an overexpression of *egl-1*, was previously observed by our team in the same condition of irradiation [11].

To go further and analyze the sperm decrease, it is important to keep in mind that while oogenesis begins at stage L4 (included in SC3) and is constitutive during adulthood, sperm production occurs only during the late L3 stage (included only in SC3 here) and is limited to ~150 spermatids per spermatheca, meaning that nematodes will produce as many eggs as their sperm number unless mated with males [70,71,72]. According to the literature, sperm number could be controlled by protein myristoylation, in which the myristic acid (C14) plays a central role to regulate the spermatogenesis to oogenesis switch [73]. Indeed, an early switch could lead to a decrease in sperm number and premature oogenesis. In our study, no significant effect was found on C14, therefore it does not seem to be responsible for sperm number decline under IR exposure. Another hypothesis could be linked with cell-cycle arrest which was previously observed [8] and reduced mitotic cell number (data not shown), therefore limiting the pool of available germ stem cells to differentiate into gametes but this hypothesis was not investigated in this work.

To sum-up, our results showed how both the germline and the soma could be involved in the regulation of lipid homeostasis, DAF-2/IIS pathway likely playing a central role. Thus, we propose a conceptual model (Figure 7) for IR response in which the germline and the soma both act to regulate lipid content by signaling opposite mechanisms. Understanding the interplay between both cell lines enabled to shed light on the different observed effects on reproduction throughout life stages. Still, the question of sperm decline remains unsolved through the study of lipids.

## 4. Materials and Methods

### 4.1. Strain and Maintenance

The Bristol N2 *C. elegans* strain and *Escherichia coli* OP50 were provided by the Caenorhabditis Genetics Center (University of Minnesota, Minneapolis, MN, USA), which is funded by NIH Office of Research Infrastructure Programs (P40 OD010440). Nematodes were cultured at 19 °C on Nematode growth medium (NGM) and M9 medium prepared by standard protocols [76]. Seeded plates with OP50 were prepared as previously described [11,27].

### 4.2. Exposure to Gamma-Radiation and Dosimetry

External gamma radiation exposure was conducted at the MIRE ^137^Cs irradiation facility at the National Institute for Radioprotection and Nuclear Safety (IRSN, Cadarache, France) as previously described [11,27]. The ^137^Cs source was provided by the CERCA LEA, Pierrelate, France.

### 4.3. Life-Stage Gamma Irradiation Experimental Design

Nematodes were first exposed in utero through a population of gravid worms previously synchronized by bleaching [77]. After 8 h of exposure of the gravid worms, eggs already laid were separated from gravid worms by a sucrose gradient (3–7%), and gravid worms were bleached to collect the eggs in utero (synchronized over 3 h). Eggs exposed in utero (~2000/replicate) were then replaced in the irradiation facility under the following three different scenarios of exposure (Figure A1):-From in utero eggs to 16h post-ovulation, covering embryogenesis (until mid-L1 stage), i.e., SC1;-From in utero eggs to 45h post-ovulation, covering early development until the beginning of meiosis (late L3 stage), i.e., SC2;-From in utero eggs to 65h post-ovulation, covering full development (until L4-YA stage), i.e., SC3.

At the end of exposure scenario SC1 and SC2, nematodes were placed in recovery until they reached the L4/YA stage. All parameters were measured when nematodes reached the L4/YA stage. Total doses are given in Appendix A.

### 4.4. Analysis of Indirect Effects of Gamma Radiation on Bacteria

To assess the effects of irradiated bacterial food on lipid metabolism, another scenario (OP50(i)) was added to the experimental design. Non-irradiated nematodes were treated as controls but grown on previously irradiated bacterial lawn. The bacteria were irradiated, in triplicates, under the same exposure condition than SC3 (i.e., 65 h at 50 mGy·h^−1^, Figure A1).

### 4.5. Assessment of Reprotoxic Effects

#### 4.5.1. Reproduction Assay

Cumulated larvae number and hatching success were quantified for each individual (20 per scenario) for eight days. Nematodes were transferred to fresh NGM petri dish every 24 h. In addition, newly laid eggs were quantified every 6 h after the transfer every day. Hatching success was analyzed by counting the newly hatched larvae 24 h after the transfer.

#### 4.5.2. Spermatids Quantification

Samples were washed three times with M9 and 15 to 25 worms per replicate were mounted on slides pre-coated with Poly-Lysine (10 µL at 0.1%). Worms were immobilized with 3 µL of 0.1 mM levamisole and dissected using a 0.45 µm gouge needle, then fixed with 2% paraformaldehyde. The freeze-cracking method was used to remove the cuticle [78]. Slides were then fixed in methanol/acetone (1:1) for 20 min at −20 °C, then rinsed three times in 1x-PBST. Slides were stained with DAPI mounting medium for 2 h at +4 °C in a dark chamber. Images were obtained with Axioobserver ZEISS Z1 microscope (Carl Zeiss AG, Oberkochen, Germany) equipped with a DAPI filter system, at 40× and 12-bit resolution and Z-stacked for each spermathecal. Spermatids were quantified using the FIJI (Fiji Is Just) ImageJ 2.1.0/1.53c software [79].

### 4.6. Assessment of Effects on Lipid Metabolism

At the end of exposure and recovery, samples were washed three times with M9 and redistributed as five replicates per condition (~1000 nematodes/replicate) for lipid and protein extraction. Samples were flash frozen in liquid nitrogen in a grinder tube with the minimal volume and stored at −80 °C until use.

#### 4.6.1. Total Lipids Extraction

Lipids were extracted with a lysis buffer (acetic acid/EDTA 1 mM) and zirconium grinding beads in a homogenizer (3 × 6500 rpm). Lipids in the aqueous phase were extracted by modified Bligh and Dyer [80]. Lipids in the organic phases were collected by evaporation under a nitrogen flow and adding of chloroform/methanol (2:1, *v*/*v*). One third of the volume was used for the analysis of neutral and polar lipids, while the remaining two thirds were used for fatty acid methyl ester (FAME) analysis.

#### 4.6.2. Analysis of Neutral and Polar Lipids

Polar and neutral lipid quantification was made by high performance–thin layer chromatograph (HPTLC) on a silica gel 60 (Merck KGaA, Darmstadt Germany). Acetone/toluene/water (91/30/8, *v*/*v*/*v*) was used as eluent for polar lipids and hexane/diethyl ether/acetic acid (17/3/0.2, *v*/*v*/*v*) for neutral lipids. Triheptadecanoin and heptadecanoic acid were used as neutral lipids standards. Polar lipids standards were phosphatidylcholine and phosphatidylethanolamine. The plates were revealed with orthophosphoric acid/copper sulfate (170 °C for 20 min) then scanned at 500 nm using a TLC Scanner 3 (CAMAG, Muttenz, Switzerland) with WinCATs software (version 1.4.4.6337) [80]. Neutral and polar lipids content was weighted with total protein content extracted with the method described below.

#### 4.6.3. Fatty Acid Methyl Esters (FAMEs) Analysis

Fatty acids were transmethylated (sulfuric acid 5% in methanol, 1 h 30, at 85 °C) into FAMEs and extracted with hexane as described in Siaut et al. (2011) [80]. Behenic acid was used as internal standard. Samples were injected in GC-MS-FID (Agilent Technologies, Inc., Santa Clara, CA, USA) with Optima-Wax (0.25 × 30 m) column (MACHEREY-NAGEL GmbH & Co. KG, Dueren, Germany) and helium as vector gaz. The samples were split for detection: MS for identification and FID for quantification. FAME content was weighted with total protein content extracted with the method described below.

#### 4.6.4. Protein Extraction and Quantification

After the separation of phases for lipid extraction, the aqueous phase was placed under nitrogen flow to evaporate residual solvents. 1% SDS was added, pH adjusted to 9 and placed under agitation for 1 h. Proteins were quantified using classical Bicinchoninic acid assay (BCA; KIT THERMO SCIENTIFIC PIERCE Illkirch-France).

#### 4.6.5. TEM Analysis

After washing, ~500 worms per replicates (3 replicates per condition) were collected and fixed with 2.5% glutaraldehyde in 0.1 M, pH 7.4 sodium cacodylate buffer for two days at 4 °C. The samples were washed for 5 min three times with the same buffer. Samples were postosmicated with 1% osmium tetroxyde in cacodylate buffer for 1 h, dehydrated through a graded ethanol series, and finally embedded in monomeric resin Epon 812. All chemicals used for histological preparation were purchased from Electron Microscopy Sciences (Hatfield, PA, USA). Ultrathin sections (70 nm; Leica-Reichert Ultracut E) were collected from different levels of each block, counterstained with 1.5% uranyl acetate in 70% ethanol and lead citrate and observed using a Tecnai F20 transmission electron microscope at 200 kV at the CoMET MRI facilities (INM, Montpellier, France). For lipid droplet analysis, 7 individuals (i.e., image) were analyzed for each condition using FIJI (Fiji Is Just) ImageJ 2.1.0/1.53c software [79].

### 4.7. Statistical Analysis

All data were obtained with a simple random sampling. Count data (spermatids count, total broodsize, hatching success, egg laying rate and lipid droplet size) were analyzed using a GLM (General Linear Model). For lipid contents (FAMEs, neutral lipids and polar lipids), when the data was normally distributed, means were compared using an Anova, otherwise, Kruskall–Wallis test was performed. Specifically, a principal component analysis (PCA) and Spearman correlation analysis were conducted on FAMEs. For each test, post-hoc analysis was conducted (Tukey and Dunnett for parametric tests, Dunn’s for non-parametric tests) and *p*-values were adjusted using the Bonferroni correction. An alpha risk of 5% was taken as significant. Statistical analysis was conducted on R Studio software, Version 1.1.423 (© 2009–2018 RStudio, Inc., Boston, MA USA), using the following packages for statistical tests: ‘car’, ’dunn.test’, ’multcomp’, ’FactoMineR’, ’factoextra’, ’psych’, ’PerformanceAnalytics’.

## 5. Conclusions

Our study aimed to understand the link between reproduction decline and lipid metabolism after exposure to IR in *C. elegans*. In order to understand precisely when these mechanisms occur, we studied chronic exposure to IR at different life stages. Our results showed that different mechanisms occur at different developmental periods (embryogenesis, early development, full development), occasioning more or less severe response in reproduction, and opposite responses in lipid metabolism. We showed that these different responses could be due to opposite signaling from the germline and the soma, contributing to maintain lipid homeostasis. Numerous studies have focused on deciphering the mechanisms behind this double response. However, it was unclear whether the lipogenic and lipolytic pathways act on the same or distinct types of lipids. In the present study, we showed that neutral and polar lipids, especially TAG and PC, are oppositely regulated after exposure to oxidative stress. In order to determine the links between these key events, we suggest that IR induce a defective germline which in turn leads to an accumulation of lipids (TAG) and favors the lipogenesis of specific fatty acids (MUFAs). In response to this misregulation, the soma acts as a negative regulator by inducing lipid catabolism (PC) to compensate this accumulation. Future studies will focus on deciphering the mechanisms behind this double response and the causes of reproduction decline.

## Figures and Tables

**Figure 1 ijms-22-10277-f001:**
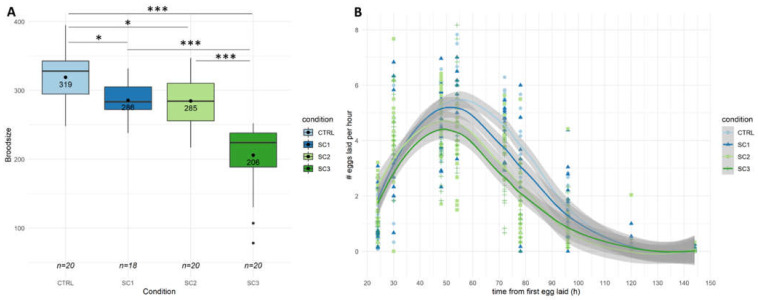
(**A**) Total broodsize during six days of egg-laying (*n* = 18–20 per condition (SC1: embryogenesis, SC2: early development, SC3: full development), Generalized Linear Model (GLM) and Tukey post-hoc with Bonferroni adjustment; ‘*’ *p*-value < 0.05; ‘***’ *p*-value < 0.001). (**B**) Egg laying rate (number of eggs laid per hour over six days) for each condition (smooth method is loess regression with 0.95 confidence interval) (significant differences were found between SC2-CTRL, SC3-CTRL, SC3-SC1, see text for *p* values).

**Figure 2 ijms-22-10277-f002:**
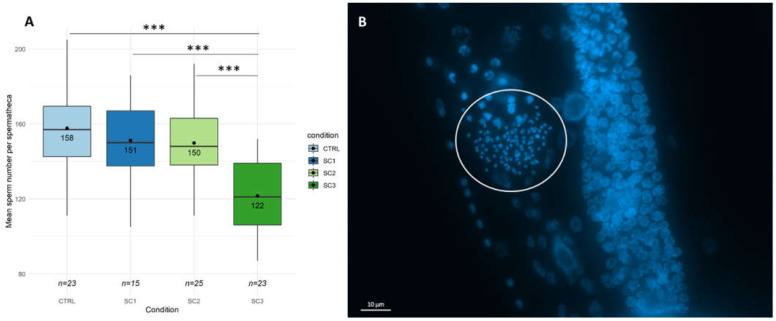
(**A**) Spermatid number per spermatheca (*n* = 15–25 per condition (SC1: embryogenesis, SC2: early development, SC3: full development), GLM, Tukey post-hoc with Bonferroni adjustment; ‘***’ *p*-value < 0.001). (**B**) DAPI-stained *C. elegans* gonad (Maximum intensity projection of Z-stacks image. DAPI staining appears in blue. White circle shows the spermatheca containing spermatid-stained nuclei counted with FIJI, on the right of the spermatheca is shown the distal gonad with proliferating germ cells).

**Figure 3 ijms-22-10277-f003:**
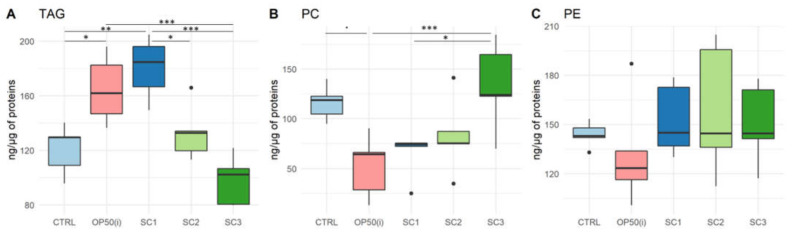
Lipid content per class (ng·µg^−1^ of total protein) for each scenario (SC1: embryogenesis, SC2: early development, SC3: full development, OP50(i): control nematodes fed on OP50 irradiated during the same time length as SC3). (**A**) Triacylglycerols (TAG) (Anova; R2 = 0.66, F(4,20) = 12.69, *p* = 2.66 × 10^−5^) (**B**) Phosphatidylcholine (PC) (Anova; R2 = 0.43, F(4,20) = 5.61, *p* = 0.0034) (**C**) Phosphatidylethanolamine (PE) (Anova; R2 = −0.06, F(4,20) = 0.67, *p* = 0.62). (All post-hoc tests are Tukey with Bonferroni adjustment; ‘*’ *p*-value < 0.05; ‘**’ *p*-value < 0.01; ‘***’ *p*-value < 0.001—*p* values from multiple comparisons are given in the text).

**Figure 4 ijms-22-10277-f004:**
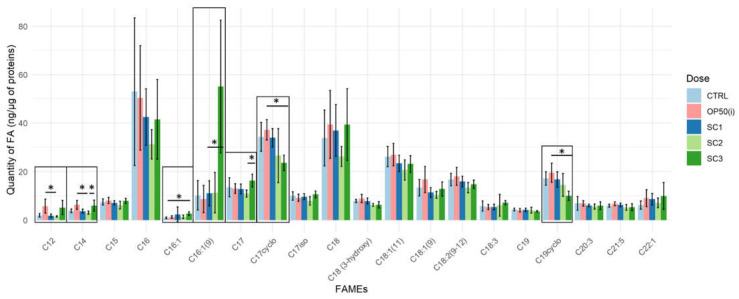
FAMEs content (ng × µg^−1^ of total protein content) for each scenario ((SC1: embryogenesis, SC2: early development, SC3: full development), Kruskal Wallis, Dunn post-hoc with Bonferroni adjustment; ‘*’ *p*-value < 0.05).

**Figure 5 ijms-22-10277-f005:**
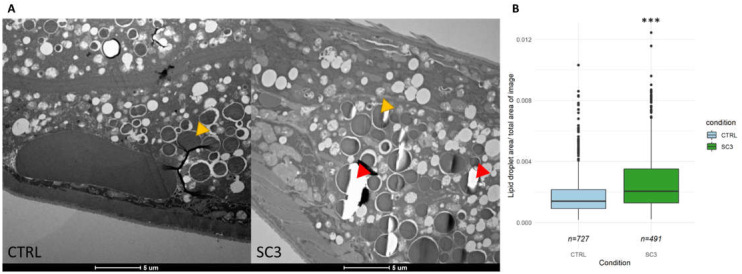
(**A**) Representative TEM images of *C. elegans* intestine (longitudinal) in each condition (Yellow arrows show the type of lipid droplets that were measured. Red arrows show tears in the slice. Images at similar zones, i.e., in the intestine and same scale were selected. See Appendix A for example of acquisition method). (**B**) Lipid droplet surface measured in TEM images (seven individuals, i.e., seven slices, were analyzed for each condition; ‘***’ *p*-value < 0.001).

**Figure 6 ijms-22-10277-f006:**
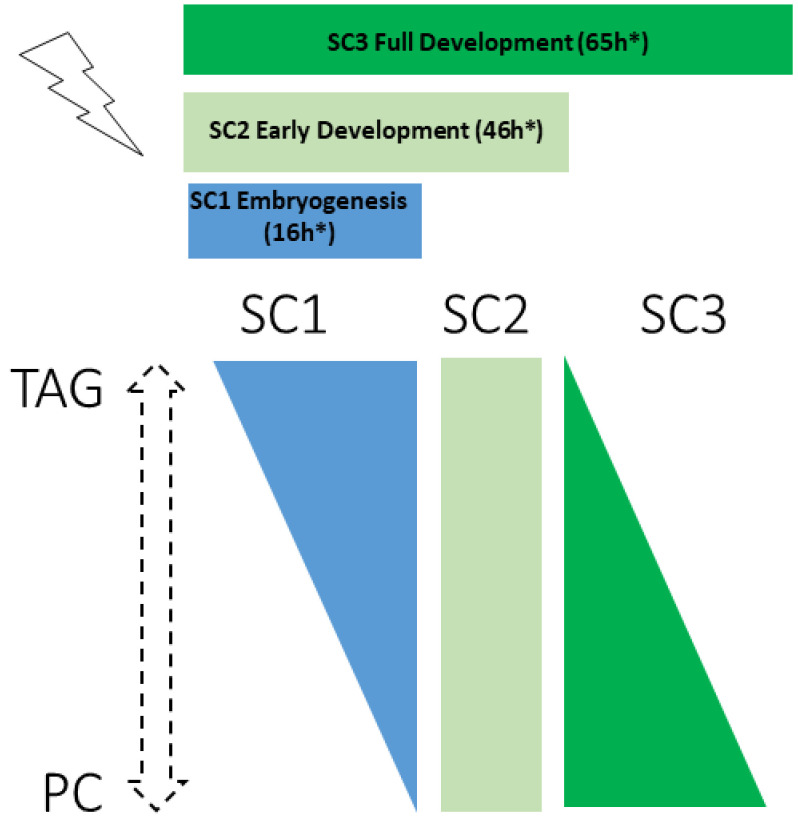
Continuous exposure to ionizing radiations induces life-stage dependent opposite regulation of PC and TAG contents.* irradiation time starting from egg stage (not including the 10 h of in utero irradiation, see Figure A1*)*.

**Figure 7 ijms-22-10277-f007:**
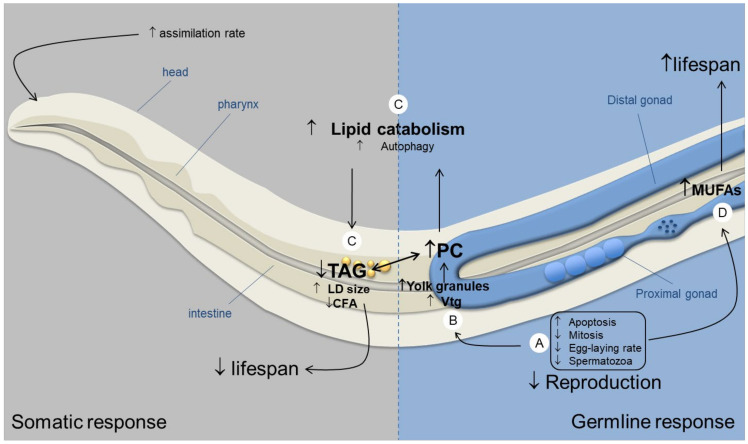
Proposed model for germline and soma regulation of lipid homeostasis after chronic exposure to IR during whole development in *C. elegans* N2 hermaphrodite (↑ corresponds to an increase, ↓ corresponds to a decrease). IR induce germline defects (**A**) with the reduced egg-laying rate, brood size and sperm number (present study), the increase of radio-induced apoptosis and cell-cycle arrest [11] and reduced mitotic cell number (results not shown). This germline deficiency and/or the stress-induced intestine autophagy [22] could lead to (**B**) an accumulation of yolk granules as shown by the increased PC content (present study) and overexpression of vitellogenins (vtg) [3]. In turn, this accumulation of fat content could lead to lipid intoxication, known to reduce lifespan [74]. In response to this accumulation, the soma acts to regulate lipid levels by (**C**) increasing autophagy and lipid catabolism (probably through the IIS pathway) as shown by the decreased TAG and CFA content (present study), and the upregulation of genes involved in autophagy after IR exposure [19]. Autophagy is known to reduce lipid intoxication, thus enhancing lifespan [75]. However, under stress conditions, previous studies have shown that lipid catabolism was associated with decreased lifespan in germline less mutants [27,35]. However, in N2 strains, longevity is not affected by IR under our conditions [8,27]. Therefore, either lipid catabolism is also detrimental to lifespan, or the germline provides an opposite signal that induces an increase of longevity, compensating the positive response of the soma, as observed through the increase of MUFA content (**D**) (present study), known to be regulated by the germline [55,57] and to enhance longevity [37,38].

## Data Availability

The data presented in this study are publicly available on request from the corresponding author. No platform already exist to upload such data.

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
