# Peer review of "Deciphering Differential Life Stage Radioinduced Reproductive Decline in Caenorhabditis elegans through Lipid Analysis"

_ijms, 2021, doi:10.3390/ijms221910277_

Round 1

Reviewer 1 Report

In their submitted manuscript Dufourcq-Sekatcheff et al. addressed the effect of the ionization radiotin on the reproductive parameters of the C. elegans among various developmental stages with the emphasis on lipid content and metabolism.

Methodology/Results

The study was well conducted and authors seeems to take special care about the proper data and statistical analysis. Here I have first minor point I would like to be addressed by  authors.

Point 1: As authors mentioned only minor portion of the lipid content of the C. elegans is synthetysed de novo when majority come from bacterial consumed by worms. Authors addressed this in their experimental design by presenting OP50 data from irradiated bacteria. Since bacterial lipid composition have such crucial effect on the resulted C. elegans lipid content I would expect that OP50 will be presented in parallel on only to Control group but for all 3 developmental periods  e.g. like SC1 control/SC1 OP50. Does authors have such data in their hands or what they expect such data can show?

Discussion 

When addressing/discussing the results authors proposed s life-stage dependent opposite regulatory model of PC and TAG contents and address some possible underlying mechanisms. My major concern here is if authors can also address the changes in cell types abundances in C. elegans with regard to their differential lipid content as summarized in my second point:

Point 2: Each cell type hase unique lipid profile (finger print) so changes in the abundancies among indicidual C. elegans cell population after exposure to IR might have significant impact on total body lipid spectrum/content without affecting lipid abundancies/metabolisms in individual cells. Does authors have in their hands some corraltive data showing the depondencies of the lipid content on depletion in individual cell population and vice versa/are able to address this point anyway? 

Overall scientific merit

According to my opinion the changes in the lipid spectrum/metabolisms will be not the major player in the reduced repoductive parameters in C. elegans after IR exposure (the complex process of meiosis would be probably the major target with potential additional effect of lipid action on it) but presented study might be interesting for C. elegans reproductive science comunity and I suggest it for publication in IJMS.

************

Line 274 - Missing reference  

Reviewer 2 Report

In this manuscript, the authors analyze the effect of gamma ionizing radiations in reproductive decline of Caenorhabditis elegans. They analyzed 3 developmental periods that could be sensitive to radiations: embryogenesis, early development, and gametogenesis. Reproductive parameters, class of lipids and fatty acids were analyzed at the L4-YA stage. They found that lipids are differently misregulated according to the considered exposure and that reproductive defects are life stage dependent, but do not fully explain radiation induced reproductive defect. Also the authors propose a conceptual model of lipid signaling after radiation stress in which both the soma and the germline participate.

-The job is well done and part of the conclusions are supported by the results obtained, but another part of the discussion is very speculative and should be reduced.

 -The authors has indicated previously that protein expression modulation is a sensitive and predictive marker of radio-induced reproductive effects, here they analyze lipids. It would have been relevant if they had analyzed proteins and lipids at the same time to increase the impact of the work.

-One question remains unsolved, what is the effect of radiation on the bacteria in the intestine.

-It is suggested to divide figure 4 in two, since due to its small size, the names cannot be read on the axes. Alternatively, increase the text size of the axes.

-line 274, error in reference.

Author Response

 We would like to thank reviewer 2 for the comments and suggestions that helped to improve the manuscript. We replied below to each of the comments (in blue characters) and we also indicated the changes we made in the manuscript. We hope that our responses and the modifications will fit with the demand.

Reviewer 2

In this manuscript, the authors analyze the effect of gamma ionizing radiations in reproductive decline of Caenorhabditis elegans. They analyzed 3 developmental periods that could be sensitive to radiations: embryogenesis, early development, and gametogenesis. Reproductive parameters, class of lipids and fatty acids were analyzed at the L4-YA stage. They found that lipids are differently misregulated according to the considered exposure and that reproductive defects are life stage dependent, but do not fully explain radiation induced reproductive defect. Also the authors propose a conceptual model of lipid signaling after radiation stress in which both the soma and the germline participate.

-The job is well done and part of the conclusions are supported by the results obtained, but another part of the discussion is very speculative and should be reduced.

We believe the reviewer refers to the last part of the discussion (i.e. part 3.3). Indeed, even though this part of the discussion relies on literature knowledge more than actual results from the present article, we believe this part is important to understand our lipid results and link them to the observed reproduction decrease. We discuss how our results and results from our previous works under the same conditions (Buisset-Goussen, Goussen et al. 2014; Kuzmic, Javot et al. 2016; Lecomte-Pradines, Hertel-Aas et al. 2017; Dubois, Lecomte et al. 2018; Dubois, Pophillat et al. 2019; Kuzmic, Galas et al. 2019) show the occurrence of mechanisms that are often found in stress response in the literature (eg, DAF-2/ IIS pathway), help to discriminate soma /germline responses and attribute the increase level of TAGs to probable soma answer (autophagy). Finally, all these elements allow to draw a map of effects and try to sort them according to time to get the set of events leading to reproductive default.

Few changes have been made in this part of discussion to better explain the goal of this 3.3 part.

 -The authors has indicated previously that protein expression modulation is a sensitive and predictive marker of radio-induced reproductive effects, here they analyze lipids. It would have been relevant if they had analyzed proteins and lipids at the same time to increase the impact of the work.

As mentioned by the reviewer, proteomic analyses have already been performed in the same SC3 design, in which the effect for reproduction is maximum (Dubois, Lecomte et al. 2018; Dubois, Pophillat et al. 2019). To complete these first works, as lipids are signaling molecules linked in C. elegans to reproductive capacity and lifespan, and partly linked to vitellogenins, already shown to be overexpressed in SC3, we focused on lipids in this work, from lipid class to FAMEs analyses. We can underline this point by adding the following sentence in the introduction lines 48-52: “Studies performed in biological media, e.g. cells, tend to show that in case of hydroxyl radicals coming from external irradiation, damage to DNA and lipids is a secondary process and that proteins are more probable initial targets, due to their relative amount and reactivity (Du and Gebicki 2004; Houee-Levin and Bobrowski 2013; Gebicki 2016). Our first works were thus dedicated to protein studies, both damage (Dubois, Lecomte et al. 2018) and expression (Dubois, Pophillat et al. 2019).”

-One question remains unsolved, what is the effect of radiation on the bacteria in the intestine.

From our point of view, two points were necessary to check to ensure the absence of bias in the study (1), and to interpret correctly the data (2). (1) To be sure that a possible change of bacteria lipid chemistry (Stark 1991) (only due to radiolysis because bacteria are dead (see mat/meth)) did not yield any reproductive effect, we added a control with irradiated bacteria (dose equivalent to SC3 dose since reproductive effects are maximum in this condition).The results of this condition were presented and discussed in the document and shown no major confounding effect with irradiation effects. Indeed, we only show that worms fed on irradiated bacteria tend to have more TAGs and less PCs lipid types (inverse trends than for irradiated worms). (2) To discriminate lipid metabolism change from a change of assimilation rate, we performed the experiment presented in the supplementary. The “effect of radiation on the bacteria in the intestine” itself, even if this is an important question, was not the main focus of our work since we focused on reprotoxic effects.

-It is suggested to divide figure 4 in two, since due to its small size, the names cannot be read on the axes. Alternatively, increase the text size of the axes.

Thank you for this check, the text size of the axes was increased.

-line 274, error in reference.

Thank you for this check, we corrected the reference.

References

Buisset-Goussen, A., B. Goussen, et al. (2014). "Effects of chronic gamma irradiation: a multigenerational study using Caenorhabditis elegans." Journal of Environmental Radioactivity 137: 190-197.

Du, J. and J. M. Gebicki (2004). "Proteins are major initial cell targets of hydroxyl free radicals." Int J Biochem Cell Biol 36(11): 2334-2343.

Dubois, C., C. Lecomte, et al. (2018). "Precoce and opposite response of proteasome activity after acute or chronic exposure of C. elegans to γ-radiation." Scientific Reports 8(1).

Dubois, C., M. Pophillat, et al. (2019). "Differential modification of the C. elegans proteome in response to acute and chronic gamma radiation: Link with reproduction decline." Sci Total Environ 676: 767-781.

Gebicki, J. M. (2016). "Oxidative stress, free radicals and protein peroxides." Arch Biochem Biophys 595: 33-39.

Houee-Levin, C. and K. Bobrowski (2013). "The use of the methods of radiolysis to explore the mechanisms of free radical modifications in proteins." J Proteomics 92: 51-62.

Kuzmic, M., S. Galas, et al. (2019). "Interplay between ionizing radiation effects and aging in C. elegans." Free Radical Biology and Medicine Volume 134: Pages 657-665.

Kuzmic, M., H. Javot, et al. (2016). "In situ visualization of carbonylation and its co-localization with proteins, lipids, DNA and RNA in Caenorhabditis elegans." Free Radic Biol Med 101: 465-474.

Lecomte-Pradines, C., T. Hertel-Aas, et al. (2017). "A dynamic energy-based model to analyze sublethal effects of chronic gamma irradiation in the nematode Caenorhabditis elegans." J Toxicol Environ Health A 80(16-18): 830-844.

Stark, G. (1991). "The effect of ionizing radiation on lipid membranes." Biochim Biophys Acta 1071(2): 103-122.
